# Training and Racing Behaviors of Omnivorous, Vegetarian, and Vegan Endurance Runners—Results from the NURMI Study (Step 1)

**DOI:** 10.3390/nu13103521

**Published:** 2021-10-07

**Authors:** Katharina Wirnitzer, Mohamad Motevalli, Derrick Tanous, Gerold Wirnitzer, Claus Leitzmann, Karl-Heinz Wagner, Thomas Rosemann, Beat Knechtle

**Affiliations:** 1Department of Research and Development in Teacher Education, University College of Teacher Education, 6020 Innsbruck, Austria; mohamad_motevali@yahoo.com (M.M.); Derrick.Tanous@student.uibk.ac.at (D.T.); 2Department of Sport Science, University of Innsbruck, 6020 Innsbruck, Austria; 3Life and Health Science Cluster Tirol, Subcluster Health/Medicine/Psychology, 6020 Innsbruck, Austria; 4Research Center Medical Humanities, Leopold-Franzens University of Innsbruck, 6020 Innsbruck, Austria; 5Faculty of Physical Education and Sports Sciences, Ferdowsi University of Mashhad, Mashhad 9177948974, Iran; 6AdventureV & Change2V, 6135 Stans, Austria; gerold@wirnitzer.at; 7Institute of Nutrition, University of Gießen, 35390 Gießen, Germany; claus@leitzmann-giessen.de; 8Department of Nutritional Sciences and Research Platform Active Ageing, University of Vienna, 1090 Vienna, Austria; karl-heinz.wagner@univie.ac.at; 9Institute of Primary Care, University of Zurich, 8000 Zurich, Switzerland; thomas.rosemann@usz.ch (T.R.); beat.knechtle@hispeed.ch (B.K.); 10Medbase St. Gallen Am Vadianplatz, 9000 St. Gallen, Switzerland

**Keywords:** running, marathon, sport, race, recreational

## Abstract

As a key modulator of training adaptations and racing performance, nutrition plays a critical role in endurance runners’ success, and the training/racing behaviors of runners are potentially affected by their diet types. The present study aimed to investigate whether distance runners with a vegan diet (i.e., devoid of foods or ingredients from animal sources), vegetarian diet (i.e., devoid of meat and flesh foods), and omnivorous diet (i.e., a mixed diet with no restriction on food sources) have different training and racing patterns in general and based on race distance subgroups. A total of 3835 recreational runners completed an online survey. Runners were assigned to dietary (omnivorous, vegetarian, and vegan) and race distance (<21 km, half-marathon, and marathon/ultra-marathon) groups. In addition to sociodemographic information, a complete profile of data sets focusing on running and racing behaviors/patterns was evaluated using a questionnaire-based epidemiological approach. There were 1272 omnivores (47% females), 598 vegetarians (64% females), and 994 vegans (65% females). Compared to vegans and vegetarians, omnivorous runners prepared for a longer time period for running events, had a higher number of half-marathons and marathons completed with a better finish time, and had more reliance on training under supervision (*p* < 0.05). The present findings indicate an important association of diet types with patterns of training and racing amongst endurance runners that may be related to different motives of omnivorous, vegetarian, and vegan runners for participating in events.

## 1. Introduction

Endurance running is a low-cost and convenient physical activity performed over various distances up to ultra-marathon at professional or recreational levels [1,2]. According to the International Association of Athletics Federation, endurance running popularity has increased by 60% over the past decade, and currently, more than 70,000 running events are held annually worldwide [3]. Training and competition are central components of the endurance runner lifestyle. Training and racing behaviors of endurance runners may vary and relies on several intra- and inter-individual factors (e.g., runners’ motives, competition level, sex, race distance, and nutrition), potentially affecting their health and performance status [4,5,6,7,8].

Pre-race preparation is crucial for runners’ success and could be considered the most important determining factor of endurance performance. Traditionally, training for an endurance event involves a high workload in the form of long endurance runs performed the days/weeks prior to competition [5,9]. However, evidence has shown that an improper increased training load will not guarantee success and could also potentially be associated with detrimental consequences negatively affecting both performance [4] and health (e.g., running-related injuries) [10]. It has been shown that endurance athletes train and compete for approximately 1000 h per year, up to 90% of which is typically performed in different forms of low-intensity exercises [11,12]. Moreover, evidence indicates that the mean training volume of marathon runners is about 50 km/week with an average training speed of 11 km/h [9,13]. Training volume seems to have a positive association with race distance, as it has been reported that half-marathoners have fewer weekly training kilometers than marathoners [14], and the highest training mileage among distance runners is held by ultra-marathoners [15]. In addition to training type and volume, the current literature available suggests that other factors including, training intensity, training frequency (or recovery duration), duration of the training season, as well as qualitative factors related to training/racing (training supervision, nutritional supports, number of racing events, tapering phase, etc.) could be listed as remarkable parameters associated with running/racing behaviors of distance runners [4,8,16,17,18].

As a key modulator of training adaptations and racing performance, nutrition plays a critical role in endurance running success [19]. During the past decades, vegan (i.e., devoid of foods or ingredients from animal sources) and vegetarian diets (i.e., devoid of meat and flesh foods) are increasingly followed for various reasons, including health, performance, ethical, and environmental concerns [20,21,22,23]. While the bio-availability of some nutrients has been reported to be lower in vegan/vegetarian than in mixed/Western diets [21,24], it has been well-documented that plant foods typically have higher amounts of carbohydrates and antioxidants [21,25,26,27], beneficially affecting endurance performance [20]. According to a recent study, about 10% of marathoners follow vegan or vegetarian diets [28], and evidence indicates that the prevalence of vegan/vegetarian diets is higher among ultra-marathoners than those who run over half- or full-marathon distances [29]. However, it has been reported that there is no difference regarding diet quality scores between runners in different race distances, but vegan and vegetarian runners had higher diet quality scores than omnivorous runners [29]. Given the well-documented interconnectedness of diet types and distance running [20,29], it seems that training and racing behaviors of endurance runners could potentially be affected by their adhered diet types.

To date, a large number of studies have been conducted on the concept of vegan/vegetarian diet and endurance performance [20,30,31,32]. Moreover, various studies have investigated the pre-race preparation and/or training behaviors of endurance runners in different race distances independently [9,33,34] or comparably [8]. However, no study has examined and compared training/racing profiles of vegan, vegetarian, and omnivorous endurance runners to date. Therefore, considering the importance of nutritional demands of endurance athletes in general, particularly of vegan/vegetarian athletes, and given the strong association between nutrition and running/racing behaviors [19], the present study was conducted to test the hypothesis whether vegan, vegetarian, and omnivorous endurance runners have different training and racing patterns.

## 2. Materials and Methods

### 2.1. Study Protocol and Ethics Approval

The present investigation is a part of the Nutrition and Running High Mileage (NURMI) Study, which has been conducted in Steps 1–3 following a cross-sectional design. The study protocol of the NURMI Study [35] was approved by the ethics board of St. Gallen, Switzerland on May 6, 2015 (EKSG 14/145) with the trial registration number ISRCTN73074080. 

### 2.2. Participants

Endurance runners were contacted and engaged mostly via social media, websites of the organizers of distance running events, running societies, email lists and runners’ magazines for health, nutrition and lifestyle, trade fairs on sports, plant-based nutrition and lifestyle, and through personal contacts. Although intended to be focused on mainly European countries, with German-speaking countries (i.e., Germany, Austria, and Switzerland) as core regions, the online investigation was spread across the globe, too, by disseminating the questionnaire of this study within the international runners’ community. Therefore, a further sample of 75 highly motivated endurance runners from non-European nations provided valuable data sets by giving accurate and useable answers. To avoid forfeiting these valuable data sets and have a bigger sample size, runners who met the inclusion criteria were included in the study to increase the representation of data provided and the consequent results. The participants’ sociodemographics and characteristics are presented in Table 1.

### 2.3. Procedures

Participants completed a short, standardized questionnaire (provided in German and English) within the NURMI Study Step 1 (preliminary study, epidemiological approach) available online at https://www.nurmi-study.com/en (accessed on 6 October 2021). The online survey started with a written description of the procedure, and runners gave their informed consent to participate in the study. Afterward, participants completed the questionnaire, which included questions about sociodemographic characteristics, adherence to a specific diet type, the distances active in running (racing and training), and running/racing behaviors. 

For successful participation in the study, four inclusion criteria were applied: (1) written informed consent, (2) no less than 18 years of age, (3) questionnaire Step 1 completed retrospectively to a race, and (4) completion of an endurance running event in the past two years and still active in running (all distances, all levels). Runners who met the aforementioned inclusion criteria were enrolled in the data analysis.

Participants were initially classified according to race distance: half-marathon and (ultra-)/marathon (data were pooled since the marathon distance is included in an ultra-marathon). The shortest and longest ultra-marathon distances reported were 50 and 160 km, respectively. Additionally, 622 highly motivated runners competing in shorter distances provided accurate answers with numerous high-quality data; however, they had not successfully participated in either a half-marathon, marathon, or ultra-marathon, but in races over distances shorter than half-marathon instead. To avoid an irreversible loss of these valuable data sets, runners who met all inclusion criteria but named races shorter than half-marathon (<21 km) race as their running event were included in the study as an additional race distance subgroup. Moreover, participants were categorized into three dietary subgroups: omnivorous (commonly known as the traditional or Western diet with no restriction on any source of food), vegetarian (no meat, including no fish/shellfish, too), and vegan diets (no products from animal sources, such as meat, fish, animal fats, milk and dairy, eggs, honey, and animal byproducts) [21,38]. 

According to the WHO [36,37], the goal for individuals should be to maintain a BMI in the range of 18.50–24.99 kg/m^2^(BMI_NORM_) to achieve optimum health. They point to an increased risk of co-morbidities with a BMI of 25.0–29.9 kg/m² and moderate to severe risk of co-morbidities with a BMI > 30 kg/m² [36,37]. Therefore, the calculated Body Mass Index (BMI_CALC_) was classified into three categories of the body weight-to-height ratio (kg/m²): ≤ 18.49 kg/m² < BMI_NORM_: 18.50–24.99 kg/m² ≥ 25 kg/m². Additionally, the BMI of active runners could be below BMI_NORM_ [39], and because some people with a higher BMI might start running to achieve and maintain stable, healthy body weight, only participants with a BMI < 30 were included in the study.

### 2.4. Data Clearance

A total number of 7422 participants took part in the survey. However, 48% dropped out, and 3835 runners completed the questionnaire. A group of 834 runners was also excluded from data analysis due to incomplete, inconsistent, and conflicting data sets. Two groups of control questions were included within different sections of the questionnaire to control for running activity and diet measures. Additionally, 156 participants without information considering running training (e.g., training time) were excluded from the study. Moreover, to control for a minimal state of health linked to a minimum level of fitness [36,37] and to further enhance the reliability of data sets, the BMI approach was implemented within data clearance. Therefore, 42 runners with a BMI ≥ 30 were excluded from data analysis. After data clearance, a group of 2864 active recreational runners (including 622 runners in < 21 km, 1032 runners in half-marathon, and 1,210 runners in marathon/ultra-marathon) with complete data sets were included in final data analysis. With regard to diet type, there were 44% (1272) omnivores, 21% (598) vegetarians, and 35% (994) vegans. As 156 runners did not report their training time, the qualitative data analysis was conducted on a group of 2708 participants. Figure 1 shows the flow diagram of participants’ enrollment in the present study. 

### 2.5. Measures

A complete profile of data sets focusing on running and racing behaviors/patterns was obtained using an epidemiological approach accomplished by the following items: nationality, age, sex, body weight, height, BMI, training volume (weekly/daily time of running), time to prepare for the major running event, training preparation, training supervision, the purpose of contribution in a running race (performance vs. joyful/enjoyment approach), participation in running events, race distance(s) completed (<21 km, half-marathon, marathon, ultra-marathon), number of completion of specific distance(s), the individual best time recorded over half-marathon or marathon distances, general strategies for competition nutrition, and adherence to different kinds of diet (omnivorous, vegetarian, vegan). Participants running on average 2.2 times per week were categorized into the low training frequency subgroup, whereas participants running on average 4.9 times per week were included in the high training frequency subgroup. Table 2 shows the sub-categorical classification of training mileage (low, medium, high) based on weekly training frequency.

### 2.6. Statistical Analysis

The statistical software R version 3.6.2 Core Team 2019 (R Foundation for Statistical Computing, Vienna, Austria) performed all statistical analyses. Exploratory analysis was performed by descriptive statistics (mean values and standard deviation (SD), median and interquartile range (IQR)). Differences between dietary subgroups and sex, training mileage (low, medium, high; e.g., daily or weekly), and running/racing behaviors were calculated using a non-parametric ANOVA. Chi-square test (χ²; nominal scale) examined the association between variables, and Kruskal–Wallis test (ordinal and metric scale) approached by using the t or F distributions with ordinary least squares, standard errors (SE), and R². Differences in weekly training by distance and diet in female and male runners are presented as effect plots (95 % confidence interval). Mosaic plots displays standardized residuals to show the relation between sex and the respective subgroups of diet type, race distance, and training volume (visualization by color: blue—high/positive numbers; red—low/negative numbers; along with brightness: dark/intense—significantly high/positive values; light—significantly low/negative values). The statistical significance level was set at *p* ≤ 0.05.

## 3. Results

A total sample of 2864 runners (57% women and 43% men) were included for statistical analysis. The median age was 37 (IQR 17, range: 18–74) years, with a median bodyweight of 66 (IQR 16, range: 40–105) kg, a median height of 1.73 (IQR 0.13, range: 1.34–2.40) m, and a median BMI of 22.0 (IQR 3.3, range: 11.4–29.9) kg/m². 84% (*n* = 2478) of participants were within the BMI_NORM_, 5% (*n* = 141) were calculated with a BMI of <18.5, and 11% (*n* = 340) with a BMI > 24.99. The countries of origin reported were Europe (97%; *n* = 2789), America (2%; *n* = 70;), and Asia (<1%; *n* = 4). 

The categorization of runners based on sex, training mileage, race distance, and dietary subgroups is shown in Table 3. The highest concentration of female runners was in the vegan group, classified into the medium training mileage subgroup, and racing at the half-marathon distance. The highest concentration of male runners was in the omnivorous group, classified into the high training subgroup, and racing at the marathon/ultra-marathon distance.

Figure 2 shows the association of “diet type” with sex and race distance subgroups. There were positive relationships between subgroups where male, omnivorous marathon/ultra-marathon runners and female, vegan < 21 km runners had the strongest relationships. A highly significant inverse relationship for female, omnivorous marathon/ultra-marathon runners was observed. 

Figure 3 displays the relationship between sex, diet type, race distance, and training mileage subgroups. As shown in Figure 3a, there is (i) a highly significant association of female omnivores racing at marathon/ultra-marathon and training at a high mileage; (ii) a highly significant association of female vegetarians racing at marathon/ultra-marathon and training at a high mileage; (iii) a significant association of female vegetarians racing at <21 km and training at a low mileage; (iv) a highly significant association of female vegans racing at <21 km and training at a low mileage; (v) a highly significant inverse association of female omnivores racing at marathon/ultra-marathon and training at a low mileage; and (vi) a highly significant inverse association of female vegans racing at marathon/ultra-marathon and training at a low mileage. Figure 3b shows (i) a highly significant association of male omnivores racing at marathon/ultra-marathon and training at a high mileage; (ii) a highly significant association of male omnivores racing at <21 km and training at a low mileage; (iii) a highly significant association of male vegetarians racing at < 21 km and training at a low mileage; (iv) a highly significant association of male vegans racing at <21 km and training at a low mileage; (v) a highly significant inverse association of male omnivores racing at marathon/ultra-marathon and training at a low mileage; and (vi) a highly significant inverse association of male omnivores racing at half-marathon and training at a high mileage.

The weekly training volume is displayed in Table 4 and Figure 4 based on diet type, sex subgroups, and their interaction. Training in kilometers per week (Figure 4a), with diet type subgroups and sex revealed a main effect of sex (F_(1, 2698)_ = 46.93, *p* < 0.001) but no main effect of diet type subgroup (F_(2, 2698)_ = 0.75, *p* < 0.473). The main effect of sex was not detected by an interaction between diet type subgroup and sex (F_(2, 2698)_ = 1.09, *p* = 0.337). Weekly training hours (Figure 4b) with diet type subgroups and sex revealed a main effect of sex (F_(1, 2698)_ = 46.94, *p* < 0.001) but no main effect of diet type subgroup (F_(2, 2698)_ = 0.75, *p* < 0.472). The main effect of sex was not detected by an interaction between diet type subgroup and sex (F_(2, 2698)_ = 1.09, *p* = 0.337).

Omnivorous runners were the subgroup most likely to train under the supervision of a professional compared to vegetarian and vegan runners (*p* < 0.001). No significant difference was found between dietary subgroups and the type of professional supervision, such as performance assessment, trainer, sports scientist, or doctor specialized in sports medicine (*p* > 0.05). Omnivorous runners were shown to train for the longest time period for endurance running events (*p* < 0.001) (Table 5).

Table 6 displays the number of completed half-marathon, marathon, and ultra-marathon events for the diet type subgroups, including the best time on average for each event. A significant difference was found between diet type subgroups and the number of completed half-marathons (*p* < 0.001). Cumulative numbers of half-marathons completed were 993 by omnivorous, 403 by vegetarian, and 643 by vegan runners. A significant difference was found between diet type subgroups and the best time to complete a half-marathon (*p* < 0.001), with the omnivorous subgroup running faster on average (107.4 ± 20.6 min). A significant difference was found between diet type subgroups and the number of completed marathons (*p* = 0.004). Omnivore runners completed a cumulative total of 590 marathons, vegetarians 226, and vegans completed 332 marathons. A significant difference was found between diet type subgroups and the best time to complete a marathon (*p* = 0.010), with the omnivorous subgroup running faster on average (224.1 ± 38.7 min). Omnivorous runners completed a cumulative total of 156 ultra-marathons, vegetarians 61, and vegans 119 ultra-marathons. No significant difference was found between dietary subgroups and either the number of completed ultra-marathons (*p* = 0.068) or the best time to complete an ultra-marathon (*p* = 0.474).

## 4. Discussion

The purpose of the NURMI Study Step 1 was to answer the colloquial gross formulated question of “who is at the start of running events?”. As a part of NURMI Study Step 1, the present study aimed to investigate and compare running and racing behaviors of a large sample of endurance runners adhering to different diet types (omnivores, vegetarians, vegans) and the potential associations with sex, training mileage, and race distance subgroups. The most important findings were (i) omnivorous runners had more reliance on training (but not nutrition) under the supervision of a professional compared to vegetarians and vegans with no between-group differences in the type of supervision, (ii) omnivorous runners reported training for a longer time in preparation for running events compared to vegetarians and vegans, (iii) there was no difference between omnivores, vegetarians, and vegans for weekly training volume (duration and mileage) in both sexes, and (iv) omnivorous runners ran a higher number of half-marathons and marathons (but not ultra-marathons) along with better times to complete them compared to vegetarian and vegan runners.

Training under the supervision of a specialist could result in beneficial advantages for the health and performance of endurance runners [40,41]. Recent evidence has shown that endurance runners participating in longer distances (i.e., marathon and ultra-marathon) benefit from receiving more professional support in different phases of training and racing compared to runners competing in shorter distances (i.e., 10 km, half-marathon) [8]. Results from the present study show that being under the supervision of a professional for training/running was more prevalent among omnivorous runners; however, a converse finding was observed regarding the diet type, where a higher number of omnivores (45%) reported to have their own nutritional strategy for running and competitions, compared to vegetarian (37%) and vegans (34%). These findings could be potentially linked to available data [22], indicating vegan/vegetarian runners have a higher level of health-consciousness than omnivorous endurance runners, who seem to be characterized by being even more performance conscious. Moreover, participants in the present study were mostly recreational runners who report having limited training schedules compared to professional runners [4]. Therefore, the small differences in the number of omnivores versus vegetarians and vegans (13% vs. 9% and 8%) training under supervision are not robust to further interpretations, and even more so while no difference between dietary groups were detected for the type of training supervision (performance assessment, trainer, sports scientist, or doctor specialized in sports medicine) in the present study. As a pivotal indicator for optimizing adaptations and improving running performance, training should be planned and conducted with great precision in different phases of overload (e.g., high volumes, great intensity, and diversity of workouts) [42]. Preparation for an endurance event with a high training volume and long endurance runs has been associated with improved performance and faster race times [43,44]. However, runners over longer distances (e.g., marathoners and ultra-marathoners) report investing greater time in training than shorter distance runners (e.g., half-marathoners) [8,14,15]. While omnivorous runners reported having a significantly longer time spent preparing for running events, the present study found that omnivorous, vegetarian, and vegan runners had similar training duration (h/week) and mileage (km/week), suggesting a null association between diet type and training volume of distance runners. Diet type could be an important indicator of daily nutrient requirements in athletic populations. Evidence indicates that elite and recreational endurance runners might not be consuming sufficient nutrients through daily foods to support their athletic needs [9,45]. Nutrient/nutritional concerns might be more critical for endurance athletes who are known to have a higher risk of low energy availability [46], such as those who follow specific kinds of diets (e.g., plant-based diets) and particularly when the diet is not appropriately planned [47]. Findings from our laboratory indicate that vegan runners have a higher intake of dietary supplements to meet their nutritional requirements [48], without sex-specific association [49]. At the same time, it has also been shown that dietary needs can be translated appropriately to a well-planned personalized diet and even ultra-endurance challenges (e.g., mountain bike race) can be successfully completed following a vegan diet [50]. It has been shown that a carbohydrate-to-protein ratio of at least 4:1 should be met in endurance athletes, and the previously reported ratio for sedentary individuals of 5:1 in vegans matches this recommendation to provide a balanced nutritional status for promoting health, performance, and recovery [20,26]. Results from the present study showed that omnivorous, vegetarian, and vegan males had a greater weekly training volume compared to their female counterparts. Interestingly, this sex-based difference in weekly training mileage was ~14.5 km/week for vegans (males: 50.06 ± 28.61 and females: 35.59 ± 20.77 km/week), while omnivores and vegetarians had lower values (~11 and ~10 km/week, respectively). This finding could be interpreted bilaterally as whether an increased capability of vegan males exists or a decreased competency for vegan females, and investigations in the future are required to examine and scrutinize this remarkable sex-based difference of training volume in vegan runners.

Irrespective of diet type, male and female runners in the present study had a lower weekly training distance compared to recreational [9] and elite [13,51] marathoners from similar studies. It has been reported that typical recreational runners complete 3.7 runs/week while elite runners complete 14.1 weekly runs on average [51]. Concerning training frequency, runners in the present study reported a lower average of training sessions per week (range: 2.22–4.93) compared to previously investigated elite (~8.1) and recreational (~4.6) distance runners [9,51]. In general, available literature shows a wide range of training volumes and frequencies reported by endurance runners [52,53], suggesting that a variety of modulating variables should be considered when interpreting the research data (and not only limited to the professionalism level). Age, for example, can be referred to particularly, while runners in the present study had a mean age of ~37 years, which could potentially alter racing and training patterns. It has also been reported that training behaviors could be affected by race distance as evidence indicates that (half-)marathoners rely more on training speed, whereas ultra-marathoners primarily rely on training volume/distance [15].

The present study revealed that omnivorous runners have a higher number of half-marathons and marathons (not ultra-marathon) and a better time to complete these two distances in comparison with vegetarians and vegans. Previous research on distance runners has focused primarily on the association between finish time and several demographic, physiological, and training parameters [18]. To the best of our knowledge, no study has focused on diet types of distance runners and the potential associations with running and racing behaviors. Running and racing motivation(s) could be considered the most important indicator of running and racing patterns/behaviors. Research indicates that all extrinsically motivated behaviors are related to outcomes independent of the activity itself, and the purposes of the activity are achievements or avoidance of negative outcomes [54]. While participation in endurance running events seems to involve both types of motivation, it has been reported that the main foundation of running is based on personal achievement, pleasure, competition, and a sense of belonging to the runners’ communities [55,56] rather than prevention of undesirable consequences. Consistently, recent evidence shows that personal goal achievement and health purposes were the strongest motivations of marathoners for participating in running events, whereas personal recognition was the weakest motivation [7,56]. Therefore, it could be assumed that motivations of recreational runners for staying active in running events, particularly their health-oriented purposes, could potentially influence their running/racing behaviors, including the number of races and finish times [1]. This consideration could be nicely matched with the present findings, as vegan/vegetarian runners are characterized by being more health conscious than omnivorous runners [22]. However, data indicate that the current COVID-19 pandemic has shifted the runners’ motivation from competition and socialization towards fitness, stress relief, and occupying time [57]. Furthermore, given the fact that the participants in the present study were recreational runners, a determinant factor affecting the number of annual races could be the selection of specific races and seasons [8], as it has been shown that most recreational runners intend to participate in running events held in spring (e.g., Boston Marathon, London Marathon) [8].

Some limitations of our study should be distinguished. In general, the NURMI Study shares with others the limitation of cross-sectional design and questionnaire-based data generation (over/under-reporting), meaning that the reliability of the data depends on the conscientiousness of participants. However, we minimized this effect using control questions that were implemented in different sections of the questionnaire. Furthermore, caution must be warranted with the interpretation of the present findings as they allow limited conclusions regarding causality. As another important limitation, the higher proportion of vegan/vegetarian populations in German-speaking countries (10–14%) compared to other Western nations might have affected the present results to some extent as 55.5% of participants in the present study were vegans or vegetarians, which is markedly higher than the worldwide prevalence as well as within German-speaking countries.

Since scientific data about endurance athletes following plant-based diets are limited, the NURMI Study aimed to enroll large numbers of participants to provide a large data set based on big sample sizes in order to allow discrimination between different diet types and to detect differences between dietary subgroups. Therefore, the present study provides valuable information of those who participate and contribute to running events, which is of special interest for coordinators of endurance running events in general but also for specialists including trainers, coaches, and nutrition professionals to guide and advise athletes involved in running while adhering to specific kinds of diet. Additionally, the present findings may help future investigations to identify specific requirements of endurance runners when adhering to vegan and vegetarian diets.

## 5. Conclusions

The present study showed that compared to vegans and vegetarians, omnivorous runners trained for a longer preparation time for running events, had more reliance on training under supervision, and had a higher number of half-marathons and marathons completed with a better finish time. However, there was no difference between omnivorous, vegetarian, and vegan runners for weekly training volume (duration and mileage) in both males and females. These findings suggest that omnivorous distance runners may have different training and racing behaviors compared to their vegan and vegetarian counterparts. As a potential explanation, different motives of omnivorous, vegetarian, and vegan runners to participate in running/racing events (e.g., health, fitness, leisure, profession, goal achievement, social reasons), as well as their state of health and/or fitness, could be considered to justify the present findings. Overall, results from the present study indicate an important association of diet types with patterns of training and racing amongst endurance/distance runners. Future research can add support by providing comparable data on patterns of training and racing amongst endurance runners, which would especially contribute to a better understanding of running/racing behaviors in vegan/vegetarian endurance athletes. 

## Figures and Tables

**Figure 1 nutrients-13-03521-f001:**
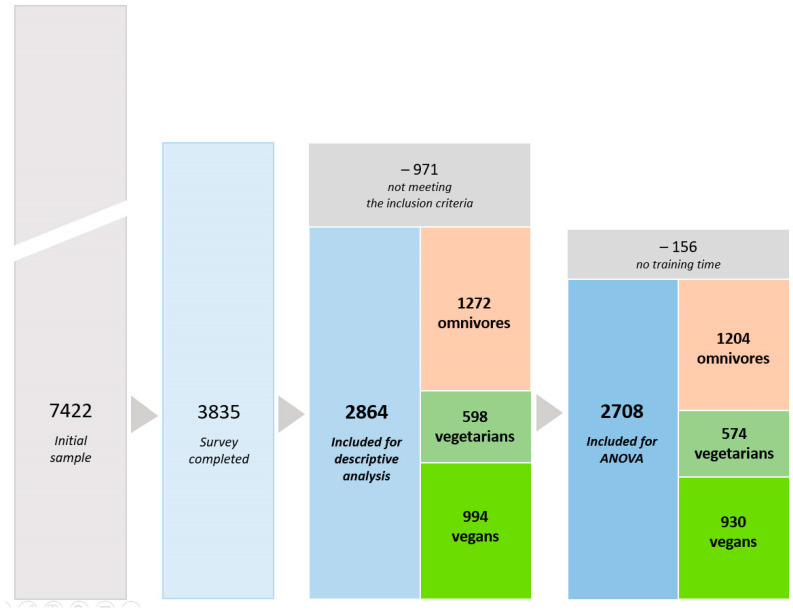
Flow of participants’ enrollment and dietary subgroups. Omnivores—have no restriction on source of food; Vegetarians—avoid all flesh foods but consume egg and/or dairy products; Vegans—avoid all foods and ingredients from animal sources.

**Figure 2 nutrients-13-03521-f002:**
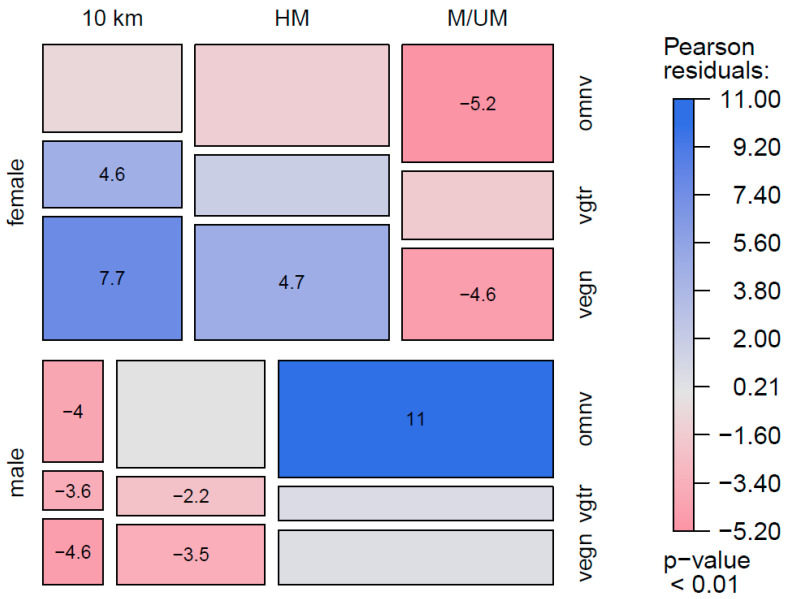
Association between Sex, Race Distance and Dietary Subgroups of Endurance runners. *Note:* Exceptionally (significantly) high values are visualized with blue and strikingly (significantly) low values are visualized with red. <21 km—less than half-marathon; HM—half-marathon; M/UM—marathon/ultra-marathon. Vegn—vegans: avoid all foods and ingredients from animal sources; vgtr—vegetarians: avoid all flesh foods but consume egg and/or dairy products; omnv—omnivores: have no restriction on source of food.

**Figure 3 nutrients-13-03521-f003:**
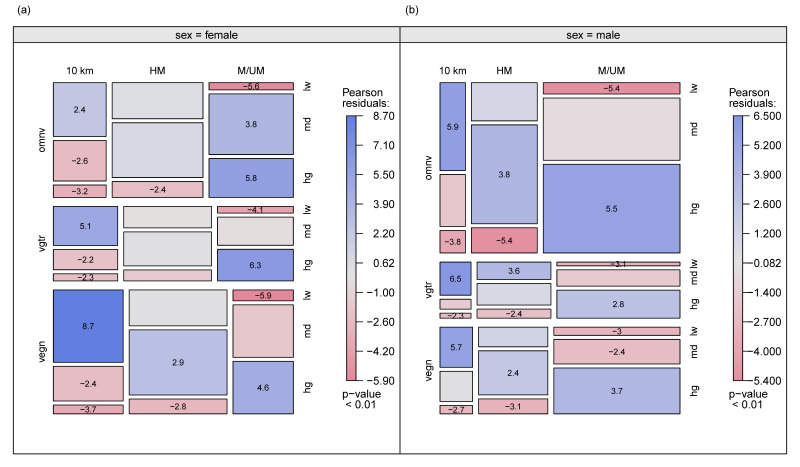
Association between Race Distance, Dietary Subgroup, and Training Mileage displayed by (**a**) Female Endurance Runners and (**b**) Male Endurance runners. *Note:* Exceptionally (significantly) high values are visualized with blue and strikingly (significantly) low values are visualized with red. <21 km—less than half-marathon; HM—half-marathon; M/UM—marathon/ultra-marathon. vegn—vegans: avoid all foods and ingredients from animal sources; vgtr—vegetarians: avoid all flesh foods but consume egg and/or dairy products; omnv—omnivores: have no restriction on source of food. Lw—low training mileage; md—medium training mileage; hg—high training mileage.

**Figure 4 nutrients-13-03521-f004:**
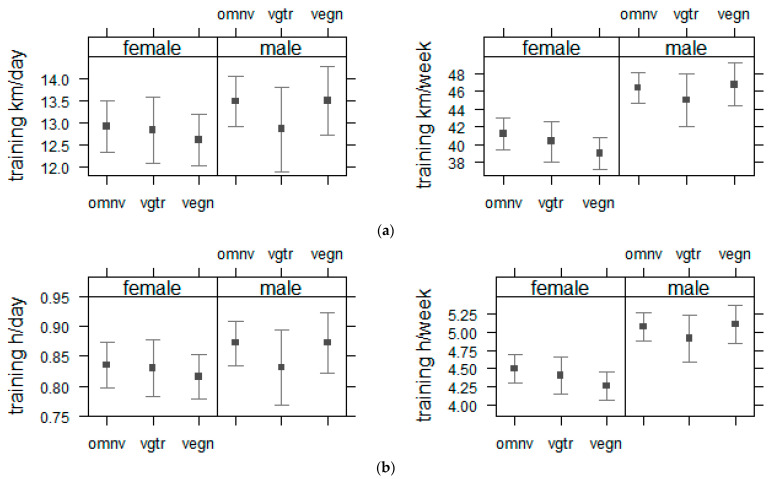
Mean effect size to display the interaction between diet type, sex, and weekly training mileage (**a**) or weekly training duration (**b**). *Note:* Results are presented as mean effect sizes with bars displaying 95%-CI with lower and upper boundaries. Omnv—omnivores: have no restriction on source of food; vgtr—vegetarians: avoid all flesh foods but consume egg and/or dairy products; vegn—vegans: avoid all foods and ingredients from animal sources. km—kilometers; h—hours.

**Table 1 nutrients-13-03521-t001:** Sociodemographic characteristics of participants displayed by dietary subgroups.

	Total	Omnivores	Vegetarians	Vegans
	*n* = 2864	*n* = 1272 (44%)	*n* = 598 (21%)	*n* = 994 (35%)
**Sex**				
Female Male	1628 (57%) 1236 (43%)	600 (47%) 672 (53%)	381 (64%) 217 (36%)	647 (65%) 347 (35%)
**Age** (years)	37 (IQR 17)	40 (IQR 17)	35 (IQR 17)	34 (IQR 15)
**Body Weight** (kg)	66.0 (IQR 16)	68.9 (IQR 16)	63.5 (IQR 14.2)	64 (IQR 14.8)
**Height** (m)	1.73 (IQR 0.13)	1.74 (IQR 0.12)	1.72 (IQR 0.13)	1.72 (IQR 0.14)
**BMI_CALC_** (kg/m²)				
18.50–25.00 <18.50 >25.00	2394 (83%) 138 (5%) 332 (12%)	1034 (81%) 32 (3%) 206 (16%)	518 (87%) 38 (6%) 42 (7%)	842 (85%) 68 (7%) 84 (8%)
**Distance Completed (total)**				
<21 km Half-Marathon Marathon/Ultra Marathon	622 (22%) 1032 (36%) 1210 (42%)	223 (18%) 435 (34%) 614 (48%)	142 (24%) 215 (36%) 241 (40%)	257 (26%) 382 (38%) 355 (36%)
**Distance Completed (females)**				
<21 km Half-Marathon Marathon/Ultra Marathon	468 (29%) 652 (40%) 508 (31%)	147 (24%) 238 (40%) 215 (36%)	113 (30%) 143 (37%) 125 (33%)	208 (32%) 271 (42%) 168 (26%)
**Distance Completed (males)**				
<21 km Half-Marathon Marathon/Ultra Marathon	154 (12%) 380 (31%) 702 (57%)	76 (11%) 197 (29%) 399 (60%)	29 (13%) 72 (33%) 116 (54%)	49 (14%) 111 (32%) 187 (54%)
**Regions/Continents**				
Europe America Asia Other	2789 (97%) 70 (2%) 4 (<1%) 1 (<1%)	1257 (99%) 13 (1%) 2 (<1%) -	592 (99%) 6 (1%) - -	940 (95%) 51 (5%) 2 (<1%) 1 (<1%)
**Nutrition and Fluid intake on race-day**				
Own strategy for running races Same as training days Same as rest days/as always I eat and drink just what I feel like	1119 (39%) 917 (32%) 224 (8%) 604 (21%)	567 (45%) 361 (28%) 91 (7%) 253 (20%)	219 (37%) 199 (33%) 49 (8%) 131 (22%)	333 (34%) 357 (36%) 84 (8%) 220 (22%)

*Note.* Results are presented as total numbers, percentage (%) and median (IQR). <21 km—less than half marathon. BMI_CALC_—Body Mass Index calculated and categorized following the WHO guidelines [36,37]. Omnivores—have no restriction on source of food; Vegetarians—avoid all flesh foods but consume egg and/or dairy products; Vegans—avoid all foods and ingredients from animal sources.

**Table 2 nutrients-13-03521-t002:** Training frequency and mileage displayed by Low, Medium, and High subgroups.

	Low	Medium	High
**Weekly Training Frequency**	2.22 (SD 0.62; 1–3)	3.35 (SD 0.59; 1–5)	4.93 (SD 1.15; 2–14)
**Kilometers per Week**	18.9 (SD 6.75; 5–54.9)	38.2 (SD 9.07; 7–82.3)	72.8 (SD 24.4; 8–220)
**Kilometers per Day**	9.48 (SD 4.38; 3.41–40)	12.6 (SD 5.21; 4–90)	17 (SD 9.76; 5–120)

*Note:* Results are presented as mean, standard deviation (SD) and range (min—max).

**Table 3 nutrients-13-03521-t003:** Sex-based comparison of dietary subgroups in different training frequencies and race distances.

	Training Mileage	Race Distance	Diet Type		
			Omnivores	Vegetarians	Vegans
**Females**	**Low**	<21 km	12% (65)	17% (60)	19% (115)
		HM	13% (75)	12% (45)	13% (80)
		M/UM	3% (14)	3% (12)	3% (15)
		total	28% (154)	32% (117)	35% (210)
	**Medium**	<21 km	9% (49)	9% (31)	9% (55)
		HM	21% (114)	20% (71)	24% (145)
		M/UM	21% (117)	14% (51)	12% (72)
		total	51% (280)	43% (153)	45% (272)
	**High**	<21 km	2% (15)	3% (11)	2% (14)
		HM	6% (33)	6% (23)	6% (33)
		M/UM	13% (75)	16% (57)	12% (72)
		total	21% (123)	25% (91)	20% (119)
	**Total**	<21 km	23% (129)	29% (102)	30% (184)
		HM	40% (222)	38% (139)	43% (258)
		M/UM	37% (206)	33% (120)	27% (159)
		total	100% (557)	100% (361)	100% (601)
**Males**	**Low**	<21 km	6% (39)	10% (20)	7% (24)
		HM	7% (44)	12% (25)	8% (25)
		M/UM	4% (27)	4% (9)	6% (19)
		total	17% (110)	26% (54)	21% (68)
	**Medium**	<21 km	4% (23)	% (6)	6% (18)
		HM	18% (114)	14% (31)	17% (56)
		M/UM	23% (149)	18% (39)	17% (57)
		total	45% (286)	35% (76)	40% (131)
	**High**	<21 km	1% (10)	1% (3)	1% (5)
		HM	4% (29)	6% (13)	6% (19)
		M/UM	33% (212)	32% (67)	32% (106)
		total	38% (251)	39% (83)	39% (130)
	**Total**	<21 km	11% (72)	14% (29)	14% (47)
		HM	29% (187)	32% (69)	31% (100)
		M/UM	60% (388)	54% (115)	55% (182)
		total	100% (647)	100% (213)	100% (329)

*Note:* Results are presented as total numbers and percentage (%), <21 km—less than half-marathon; HM—half-marathon; M/UM—marathon/ultra-marathon. Omnivores—have no restriction on source of food; Vegetarians—avoid all flesh foods but consume egg and/or dairy products; Vegans—avoid all foods and ingredients from animal sources.

**Table 4 nutrients-13-03521-t004:** Weekly mileage and duration of training in female and male runners, and their interaction with diet type subgroups.

	Omnivores	Vegetarians	Vegans
**Female**	*n* = 557	*n* = 361	*n* = 601
Weekly mileage in km	40 (22.16)	38.18 (22.96)	35.59 (20.77)
CI (Lower-Upper boundary)	41.17 (39.32–43.01)	40.33 (38.03–42.63)	39.01 (37.19–40.83)
Weekly duration in h	4.37 (2.42)	4.17 (2.51)	3.89 (2.27)
CI (Lower-Upper boundary)	4.50 (4.30–4.70)	4.41 (4.16–4.66)	4.26 (4.07–4.46)
**Male**	*n* = 647	*n* = 213	*n* = 329
Weekly mileage in km	51.03 (27.45)	48.05 (28.01)	50.06 (28.61)
CI (Lower-Upper boundary)	46.41 (44.64–48.18)	44.99 (41.99–47.98)	46.76 (44.35–49.18)
Weekly duration in h	5.58 (3)	5.25 (3.06)	5.47 (3.13)
CI (Lower-Upper boundary)	5.07 (4.88–5.27)	4.92 (4.59–5.25)	5.11 (4.85–5.38)

*Note:* Results are presented as total numbers (*n*), mean and standard deviation (SD), 95%-CI with lower and upper boundaries; km—kilometers; h—hours. Omnivores—have no restriction on source of food; Vegetarians—avoid all flesh foods but consume egg and/or dairy products; Vegans—avoid all foods and ingredients from animal sources.

**Table 5 nutrients-13-03521-t005:** Training type and time period displayed by dietary subgroup.

	Omnivores (*n* = 1272)	Vegetarians (*n* = 598)	Vegans (*n* = 994)	Statistics
**Training Preparation**				
Under the direction of a professional	13% (169)	9% (51)	8% (75)	χ^2^_(2)_ = 22.47; *p* < 0.001
Alone and independently	87% (1103)	91% (547)	92% (919)	
**Professional Supervision**				
Performance assessment	36% (60)	20% (10)	34% (26)	χ^2^_(2)_ = 4.66; *p* = 0.097
Trainer	89% (151)	96% (49)	96% (73)	χ^2^_(2)_ = 4.56; *p* = 0.102
Sports scientist	12% (21)	8% (4)	13% (10)	χ^2^_(2)_ = 0.96; *p* = 0.618
Doctor specializing in sports medicine	11% (19)	10% (5)	7% (5)	χ^2^_(2)_ = 1.29; *p* = 0.525
**Training Period**				
1–2 months	23% (297)	29% (171)	31% (313)	H_(2)_ = 26.77; *p* < 0.001
3–4 months	47% (594)	45% (268)	46% (460)	
4–6 months	20% (251)	19% (111)	15% (152)	
7–8 months	4% (57)	4% (25)	3% (28)	
9–10 months	3% (38)	1% (8)	2% (18)	
More than a year	3% (35)	3% (15)	2% (23)	

*Note:* Results Results are presented as total numbers and percentage (%). Omnivores—have no restriction on source of food; Vegetarians—avoid all flesh foods but consume egg and/or dairy products; Vegans—avoid all foods and ingredients from animal sources.

**Table 6 nutrients-13-03521-t006:** Total numbers of completed half-marathon, marathon, and ultra-marathon events. The best time is displayed by dietary subgroups in minutes (mean ± standard deviation).

	Omnivores	Vegetarians	Vegans	Statistics
**Half-Marathon**				
Completed Events	933	403	643	H_(2)_ = 45.32; *p* < 0.001
*1*	151 (16%)	92 (23%)	153 (24%)	
*2*	142 (15%)	74 (18%)	150 (23%)	
*3–4*	201 (22%)	82 (20%)	133 (21%)	
*5–7*	165 (18%)	57 (14%)	92 (14%)	
*>7*	271 (29%)	97 (24%)	114 (18%)	
*undetermined*	3 (<1%)	1 (<1%)	1 (<1%)	
Best Time (min)	107.4 (SD 20.6)	112.2 (SD 20.5)	113.6 (SD 22.1)	F_(2, 1975)_ = 21.53; *p* < 0.001
**Marathon**				
Completed Events	590	226	332	H_(2)_ = 10.94; *p* = 0.004
*1*	148 (25%)	61 (27%)	96 (29%)	
*2*	104 (18%)	32 (14%)	72 (22%)	
*3* *–* *4*	103 (17%)	49 (22%)	78 (23%)	
*5* *–* *7*	86 (15%)	32 (14%)	34 (10%)	
*>7*	145 (25%)	52 (23%)	51 (15%)	
*undetermined*	4 (<1%)	-	1 (<1)	
Best Time (min)	224.1 (SD 38.7)	232.1 (SD 39.2)	231.4 (SD 41.8)	F_(2, 1145)_ = 4.63; *p* = 0.010
**Ultra-Marathon**				
Completed Events	156	61	119	H_(2)_ =5.37; *p* = 0.068
*1*	32 (21%)	13 (21%)	30 (25%)	
*2*	15 (10%)	15 (25%)	26 (22%)	
*3* *–* *4*	32 (21%)	5 (8%)	27 (23%)	
*5* *–* *7*	18 (12%)	10 (16%)	19 (16%)	
*>7*	26 (17%)	9 (15%)	5 (4%)	
*undetermined*	33 (21%)	9 (15%)	12 (10%)	
Best Time (min)	735.1 (SD 195.1)	769.4 (SD 192.5)	759.8 (SD 212.9)	F_(2, 279)_ = 0.75; *p* = 0.474

*Note:* Results are presented as total numbers, percentage (%), mean, and standard deviation (SD). Omnivores—have no restriction on source of food; Vegetarians—avoid all flesh foods but consume egg and/or dairy products; Vegans—avoid all foods and ingredients from animal sources.

## Data Availability

The data sets generated during and/or analyzed during the current study are not publicly available but may be made available upon reasonable request. Subjects will receive a brief summary of the results of the NURMI Study, if desired.

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
