# Peer review of "Training and Racing Behaviors of Omnivorous, Vegetarian, and Vegan Endurance Runners—Results from the NURMI Study (Step 1)"

_nutrients, 2021, doi:10.3390/nu13103521_

Round 1

Reviewer 1 Report

This is well written paper that is easy to read. Based on the aims of the study, nature of the data collected and statistical analysis performed, the conclusions are sound, and not overstated. Limitations are appropriately addressed.

Author Response

Dear Reviewer #1,

Thank you for your kind and positive feedback, and the opportunity to revise the manuscript Nutrients-1379463 entitled “Training and Racing Behaviors of Omnivorous, Vegetarian, and Vegan Endurance Runners – Results from the NURMI Study (Step 1)”.

We appreciate much the statement of Reviewer #1, thank you very much!

Kind regards,

Katharina Wirnitzer, Beat Knechtle, and the team of authors

Reviewer 2 Report

Thank you for an excellent study that is very well described! I only have one recommended change to the paper. The study has as a key element the diet types of the participants: omnivore, vegetarian and vegan. These diet types are listed in the title, abstract, and introduction. In addition, the data are in columns based on diet type in Table 1 . How your study defined these diet types, however, is not presented until line 139 (page 4). People may define themselves as consuming one of these categories of diet - but their concept of these diet definitions may not match how your study has defined these diets. It is very important that you define these diet categories when you first use them in  the abstract. I recognize that this will take up significant abstract space but  readers cannot  understand even the introduction accurately without this information. In addition, to avoid possible confusion for the reader, I also recommend that these terms be defined in footnotes of all tables and figures where the terms are used.

Author Response

Dear Reviewers #2,

Thank you for your kind and positive feedback, and the opportunity to revise the manuscript Nutrients-1379463 entitled “Training and Racing Behaviors of Omnivorous, Vegetarian, and Vegan Endurance Runners – Results from the NURMI Study (Step 1)”.

We would like to thank Reviewer #2 for this valuable and constructive comment:

“Thank you for an excellent study that is very well described! I only have one recommended change to the paper. The study has as a key element the diet types of the participants: omnivore, vegetarian and vegan. These diet types are listed in the title, abstract, and introduction. In addition, the data are in columns based on diet type in Table 1. How your study defined these diet types, however, is not presented until line 139 (page 4). People may define themselves as consuming one of these categories of diet - but their concept of these diet definitions may not match how your study has defined these diets. It is very important that you define these diet categories when you first use them in the abstract. I recognize that this will take up significant abstract space but readers cannot understand even the introduction accurately without this information. In addition, to avoid possible confusion for the reader, I also recommend that these terms be defined in footnotes of all tables and figures where the terms are used.”

In response to this appreciated comment, we revised the different parts of the manuscript accordingly, which has led to a significant improvement of the paper. Please see our revisions in lines:

23-26 (abstract), 75-77 (introduction), 108-118, 163-165, 170-190, 203-205, 346/350-356/360, 364-370, 398-409 (methods), 411-417, 438-439, 449-451, 474, 489, 495-498, 620-621, 627, 629-631, 650-652, 679-681 (results), 699-700, 730, 785, 787-788, 818-821 (discussion), as well as 852-855.

 Please note: at the turn of page 4 to 5 as well as page 9 to 10 there is each a huge increment of line number from 206 up to 350, and 499 up to 629 (maybe due to formatting figure/table!

We hope our revisions satisfy the existing concerns of the editors and reviewers. Changes in the manuscript have been highlighted via “track changes”.

Kind regards,

Katharina Wirnitzer, Beat Knechtle, and the team of authors